# Molecular Breeding to Overcome Biotic Stresses in Soybean: Update

**DOI:** 10.3390/plants11151967

**Published:** 2022-07-28

**Authors:** Niraj Tripathi, Manoj Kumar Tripathi, Sushma Tiwari, Devendra K. Payasi

**Affiliations:** 1Directorate of Research Services, Jawaharlal Nehru Krishi Vishwa Vidyalaya, Jabalpur 482004, India; 2Department of Molecular Biology and Biotechnology, Rajmata Vijayaraje Scindia Krishi Vishwa Vidyalaya, Gwalior 474002, India; drmanojtripathi64@gmail.com (M.K.T.); sushma2540@gmail.com (S.T.); 3Regional Agricultural Research Station, Sagar 470002, India; dpayasi@gmail.com

**Keywords:** molecular markers, crop improvement, resistance, linkage groups, pathogens

## Abstract

Soybean (*Glycine max* (L.) Merr.) is an important leguminous crop and biotic stresses are a global concern for soybean growers. In recent decades, significant development has been carried outtowards identification of the diseases caused by pathogens, sources of resistance and determination of loci conferring resistance to different diseases on linkage maps of soybean. Host-plant resistance is generally accepted as the bestsolution because of its role in the management of environmental and economic conditions of farmers owing to low input in terms of chemicals. The main objectives of soybean crop improvement are based on the identification of sources of resistance or tolerance against various biotic as well as abiotic stresses and utilization of these sources for further hybridization and transgenic processes for development of new cultivars for stress management. The focus of the present review is to summarize genetic aspects of various diseases caused by pathogens in soybean and molecular breeding research work conducted to date.

## 1. Introduction

Soybean is a well-known leguminous crop with a high percentage of protein and oil in seed [1]. Despite having its origin in China [2], it is extensively cultivated in most parts of the world [3]. Due to its adaptability to different climatic zones, soybean gained popularity and became one of the top crops, i.e., wheat, paddy and maize (http://faostat.fao.org/, accessed on 1 July 2022). Soybean is employed in the production of various food as well as industrial products. Apart from these applications, it is also used as animal feed [4].

An array of biotic [5,6] and abiotic [7,8,9,10,11] factors are responsible for yield reduction in soybean. Most of the abiotic factors depend upon climatic conditions [12]. Nevertheless, biotic factors involve pathogens such as bacteria, viruses, fungi, nematodes, etc. [13]. Among the diseases reported in soybean, 29 are fungal, 6 bacterial, 18 viral, 6 nematodal and 3 mycoplasmal [14]. Amongst all fungal diseases, about 10 have a consistent presence in diverse parts of the world. Of these, six pathogens, viz., *Sclerotium rolfsii*, *Macrophomina phaseolina*, *Colletotrichum truncatum*, *Phakopsora pachyrhizi*, *Cercospora sojina* and *Cercospora kikuchii*, are severe in India. Even though a lot of control measures have been developed and adopted, a combined approach for management of these pathogens is stilllacking [15]. Diseases caused by these pathogens are generally controlled by means of chemicals [16]. Controlling diseases with the application of chemicals is costly at the farmer level. It also causes environmental and water pollution because of the dangers associated with these chemicals [17]. Development of resistant varieties reduces the use of chemicals [18], andalso helps in the reduction in environmental pollution and providing safe food to humans [19].

Identification or development of resistant soybean genotypes against different diseases is a major challenge [20]. Molecular breeding technologyhas proven its efficiency in the transfer of genes toa desired cultivar. These tools are less time consuming in comparison to traditional plant breeding [21] as desired plants can be selected at the initial stage of their growth. Identification of quantitative trait loci (QTLs) provides basic information about the association of a specific molecular marker with desired trait [22]. It also defines the distance between a flanking marker and gene of interest. Resistance in soybean has been reported to be mono or polygenic. The present review provides an insight into molecular breeding approaches adopted in the identification of biotic stressresistance genes and markers associated with the targeted genes in soybean.

## 2. Soybean Diseases and Molecular Developments

### 2.1. Fungal Diseases

#### 2.1.1. Soybean Rust

A potentially devastating foliar disease instigated by two meticulously associated obligate fungal species, *Phakopsora pachyrhizi* Sydows and *P. meibomiae* (Arthur), is posing a serious threat to soybean cultivation [23] in the southern hemisphere, mainly in Asia (Taiwan, Thailand, Japan and India), Africa (South Africa) and South America (Paraguay, Brazil and Argentina). Major challenges to manage soybean rust disease are the special aptitude of *P. pachyrhizi* to cause infection in an extensive range of crop species, i.e., 95 species from 42 genera of the family Papillionaceae [24]. This disease has variable effects on soybean yield as it may cause 80% yield loss in the regions favorable for growth and multiplication of the causal organism [25].

An initial study conducted to elucidate the genetic basis showed that resistance to rust is dominant over susceptibility [26]. There are six independently inherited dominant resistance genes, viz., *Rpp*1 [27,28,29] identified in PI 200,492 [27], *Rpp*2 [28] from PI 230,970 [30], *Rpp*3 [30,31] from PI 462,312 [32], *Rpp*4 from PI 459,025 [33], *Rpp*5 [34] and *Rpp*6 from PI 567102B [35]. Recessive R genes [36] and *Rpp*1b [37] for resistance against rust in soybean have also been reported. According to Langenbach et al. [23], use of recessive *R* genes for development of SBR-resistant cultivars may be a better approach. Earlier, three recessive *R* genes were recognized in different soybean genotypes, i.e., PI 200456, PI 224,270 and BR01-18437 [36,38]. Exploitation of these R genes for the development of SBR-resistant cultivars through breeding and genetic engineering approaches is still awaited [39].

More than 112 genes have been detected during transcriptomic analysis after infection with *P. pachyrhizi* in host plants [40]. The gene *Rpp*1 confers an immune response to Asian soybean rust. However, visible signs were not seen in the plants with this gene [41]. The resistant response arbitrated by genes *Rpp*2 to *Rpp*5 loci associatedfungal growth with an oversensitive corresponding response [34,42]. Resistance in F_2_ of crosses involving accessions PI587886 and PI587880A has been segregated in 1:2:1 (resistant:segregating:susceptible) as a single gene other than *Rpp*1 as a source of resistance. Appearance of reddish-brown lesions exclusively in heterozygous conditions confirms the incidence of incomplete dominance [43]. Calvo et al. [36] demonstrated that these genes are non-allelic and resistance is governed by a recessive allele that may be used to develop more durable resistance. The appropriate use of resistance genes could avert Asian soybean rust development. The use of resistance gene to inhibit the growth of fungus is an imperative aspect of a resistant genotype [44].

In studies on three lines of soybean, one genomic region has been detected with SSR molecular marker BARC-Sct_187 on linkage group G where L85-2378 shared an allele with PI 200,492 and was polymorphic with cultivar Williams 82. Highly significant independent assortment between the *Rpp*1 gene and Sct_187 indicates tight linkage between the two loci [45]. Susceptible reaction of genotype PI200492 and Tainung 3 to Australian isolate (Q-2) was found, while resistance by genotype Tainung 4 showed the presence of different resistance gene loci [45]. The contiguous markers Sct_187 and Sat_064 are tightly linked to the *Rpp*1 locus with variation estimates of 0.46 and 0.84, respectively, which indicates high polymorphism of the SSR markers in a wide range of crosses [46].

SSR molecular markers linked to rust resistance have been documented in cultivar FT 2 in the linkage group (LG) C2 [47] of the map testified by Cregan et al. [46]. A resistance gene from the cultivar Hyuuga was mapped to a ~3cM interval on LG-C2 between Satt134 and Satt460 [48], whereas *Rpp*3 was also mapped at the same interval [45]. The *Rpp*1 locus has been mapped to a ~1cM interval on LG-G between Sct_187 and Sat_064 [45]. Meanwhile, the *Rpp*4 locus was mapped on chromosome 18 on linkage group G within 1.9 cM [49] and within 2.8 cM [34] of SSR marker Satt288. Sequencing of reverse transcription polymerase chain reaction outcomes exposed that *Rpp*4C4 (PI 459025B) was extremely expressed in the resistant genotype, whereas expression of the other intrant genes was nearly untraceable. This supports *Rpp*4C4 (PI 459025B) as the sole candidate gene for *Rpp*4C4-mediated resistance to rust [50]. In a recent study, molecular approaches were employed for improvement of soybean rust resistance in Vietnamese elite soybean [51]. Molecular markers were efficaciously used in a backcross breeding scheme to incorporate the *Rpp*5 gene of SBR resistance into HL203, a leading Vietnamese soybean variety, from two contributor lines, DT2000 and Stuart 99084B-28. In this study, the *Rpp*5 locus was found situated in the N linkage group between markers Sat_275 and Sat_280. Based on the molecular data, Maphosa et al. [52] stated effective pyramiding of three resistance genes, namely, *Rpp*2, *Rpp*3 and *Rpp*4, in pair-wise combinations in the F_2_ generation of soybean.

According to a recent review published by Chander et al. [1], seven loci (*Rpp*1 to *Rpp*6) have been identified to date with varying degrees of resistance. A virus-induced gene silencing method is being applied in the molecular detection of resistance in plants. Meyer Jenelle et al. [50] used this technology and identified resistant accession PI459025B against *P. pachyrhizi* in soybean. Further, in a recent study Pedley et al. [53] also used this technology for characterization of *Rpp*1. In their study, *Rpp*1 was found to be located on chromosome 18 between Sct_187 and Sat_064 molecular markers. According to their findings, it is recognized that the *Rpp*1 gene is a unique gene among the all Rpp genes investigated, as it confers an immune response (IR) to avirulent *P. pachyrhizi* isolates. A gene silencing experiment revealed that *Rpp*1 is a ULP1-NBSLRR protein and plays a key role in the immune response.

Fungicide application is a common strategy to mitigate SBR but it increases the production cost [54]. However, adopting SBR-resistant cultivars may reduce this cost [55]. Development of SBR-resistant soybean varieties is a challenging task [56]. For identification of sources of resistance gene/s, it is important to screen a wide range of soybean germplasm lines. Despite the identification of seven genes/loci against various specific pathotypes, they seem ineffective after infection with other pathotypes. In this situation, the importance of gene pyramiding increases. Pyramiding of identified *Rpp* genes in a single soybean genotype may provide resistance against various pathotypes of *P. pachyrhizi.* The combination of multiple genes in a single genotype may increase resistance due to cumulative effects of combined genes. Reports are available on application of gene pyramiding in soybean to develop resistance against soybean rust disease. *Rpp*2, *Rpp*4 and *Rpp*5 were combined in a single soybean genotype and revealed higher resistance against SBR [57]. In a similar way, the gene pyramiding approach was also used to combine *Rpp*2, *Rpp*3 and *Rpp*4 with cumulative resistance [53]. These reports are clear reflections of the gene pyramiding approach to increase disease resistance in soybean crop [1]. Recent research on applications of marker-assisted selection in combination with line breeding was found to be useful in developing soybean cultivars containing ASRresistance genes. It helped in the release of two new soybean cultivars, viz., JFNC 1 and JFNC 2, in Paraguay. Both of the cultivars consisted of three ASR resistance genes, namely: *Rpp*2, *Rpp*4 and *Rpp*5 [58].

#### 2.1.2. Rhizoctonia Root Rot

A predominantly soilborne fungal disease triggered by *Rhizoctonia solani* Küuhn (teleomorph *Thanatephorus cucumeris* (Frank) Donk) results in 60–70% yield losses in India, 30 to 60% in Brazil and 30 to 45% in the United States [59]. Chemical, biological and cultural control of *Rhizoctonia* root rot, a major soilborne disease of soybean, is difficult due to its wide host range [60]. Resistance to *Rhizoctonia* root rot has been reported to be quantitatively inherited with additive gene action of two to three genes [61,62]. Moreover, the inability to categorize disease indexing into separate classes of resistance and susceptibility in the segregating population is an indication thatresistance is a quantitative attribute. The cumulative gene effects showthat it is possible to breed a cultivar with a higher level of resistance by pyramiding resistance genes [63]. The soybean accession PI 442031 consistentlyshowed abstemious resistance to *Rhizoctonia* root rot instigated by an array of isolates of anastomosis assemblage (AG-4), the greatest communal group isolated from diseased soybean plants [64].

Substantial correlationbetween SSR molecular markers and resistance to *Rhizoctonia* rot was distinguished by the solo factor investigation of variance with marker genotypic classes as the explanatory variable and the disease score as the response variable [65]. A study on inheritance of resistance in moderately resistant soybean cultivar PI 442031 and four moderately susceptible commercial cultivars with significantly associated SSR molecular markers Satt281, Satt177 and Satt245 showed that marker-assisted selection coupled with phenotypical selection in later generations facilitated the development of soybean genotype resistance to *Rhizoctonia* root rot [66]. SSR marker Satt177 located on linkage group A2, Satt281 on linkage group C2 and Satt245 on linkage group M were found linked with resistance to *Rhizoctonia* root rot [67], a trait controlled by both major and minor genes [68].

Screening against root rot resistance performed with the help of these three SSR markers showed the allelic variation for this trait [69,70]. All three markers amplified five alleles each. These markers were highly polymorphic with a high PIC value (Satt177: 0.684; Satt281: 0.608 and Satt245: 0.575), and also amplified rare alleles. The cluster analysis grouped the genotypes into two major groups [71]. Even these markers are closely associated with QTLs which together are responsible for 49% of the phenotypic disparity and 73% of the genotypic dissimilarity [65]. Earlier, Dorrance and Mills [69] reported varietal resistance to *R. solani* in some soybean lines. Further, Sserunkuma [70] identified a significant QTL on Chr 10 (LG-O) that accounted for 43.1% of the discrepancy against *R. solani*. This QTL was reported to be located in the same section as QTLs described in response to *Sclerotinia sclerotiorum* and *Phytophthora* spp. fungal pathogens. Recently, Upadhyay et al. [71] used the published SSR markers for validation in Indian soybean genotypes against *Rhizoctonia* root rot. They identified some of the diverse lines for future breeding purposes.

Biochemical indicators were analyzed to determine *Rhizoctonia* root rot resistance in soybean and significant correlation between the biochemical indicator and resistance was observed [72]. According to the authors, increased level of phenols, soluble sugar and other biochemical indicators have a role in development of resistance in soybean plants.

#### 2.1.3. Charcoal Rot

Charcoal rot is triggered by *Macrophomina phaseolina*, a soil borne fungus. It is also responsible for yield reduction in soybean. According to Luna et al. [73], charcoal rot is a significant but misidentified disease. This disease was first reported in the year 1949 in the USA. The infection caused by *M. phaseolina* in crops may be due to the presence of two toxins, *phaseolina* and (−)-*botryodiplodin* [74,75,76]. However, further research is needed to confirm the efficiency of these toxins as causative agents of charcoal rot disease [75].

Favorable environmental conditions for growth and multiplication of *M. phaseolina* in plants provide a way to start infection in the vascular system. In the second step, it interferes with the transportation of water and nutrients towards leaves, which causes symptoms of disease and further death of pre-mature leaves [77,78]. After harvesting of the crop, microsclerotia return to the soil and survive for at least two years [79,80].

Management of charcoal rot disease includes, first, options for applications of fungicides and, second, biological control. Nevertheless, these tactics have not been found effective to deliver complete control [81]. Under such circumstances, genetic resistance may play a major role to manage charcoal rot [82]. Only partial resistance against *M. phaseolina* has been shownin soybean crops [83,84,85,86]. Breeding in the direction of progress of charcoal rot-resistant soybean cultivars is difficult owing to polygenic inheritance [87].

In a recent study conducted by da Silva et al. [88], a total of 140 F_2_-derived lines from a population were developed by crossing PI 567562A (resistant) and PI 567,437 (susceptible) genotypes for identification of QTL/s linked with charcoal rot resistance in soybean. The plant population was genotyped with 5403 SNP markers covering 20 chromosomes. One QTL was identified on chromosome 15 and two on chromosome 16.

#### 2.1.4. Fusarium Wilt (Sudden Death Syndrome, SDS)

This disease is instigated by the soilborne fungi *Fusarium solani* (Mart.) Sacc. F.sp. *glycines* (*Burk.*) Snyd. & Hans and *Fusarium virguliforme* (Aoki, O’Donnel, Homma & Lattanzi), with infection strictly restricted to roots. Yield losses up to 5–15% have been reported in the USA due to SDS [89]. It is an important disease distressing major soybean producing nations, viz., USA, Argentina, Brazil and Thailand [90]. Plants infected with *Fusarium solani* display leaf chlorosis and necrosis, and under severe infection leaves drop off with abortion of pods in the late reproductive stage. Sudden death syndrome resistance is tough to regulate under field conditions as disease appearance is frequently environmentally sensitive, erratic and irregular and treatment is time intense and expensive [91,92,93,94]. Resistance to SDS is fractional and the constancy and yield compatibility of partial disease resistance delivers benefits over complete resistance [95,96].

Its genetics are complex because of multi-genetic control [97,98] and the environmentally influenced nature of the disease [99]. Its resistance is controlled by quantitative loci. Resistance to *Fusarium* wilt is instigated by four genes in a cluster with two duos in near linkage or by a two-gene cluster with each gene exhibiting pleiotropy [100]. Resistance to SDS may be controlled by single dominant gene *Rfs*1 under greenhouse conditions [101]; a small number of major QTLs regulate partial resistance [102]; a number of QTLs [94,95] act as a qualitative locus. Candidate genes were recognized for two loci. *QRfs*1 providesresistance to root infection and *QRfs*2 provides resistance to leaf scorch. Ref. [103] testified QTLs conferring nine LGs (A2, C2, D2, F, G, I, J, L and N) for resistance to root infection (Rfs1) on linkage group G. Soybase [104] includes more than 56 recordsof QTLs for *Fusarium* wilt in soybean.

Resistance to *Fusarium* wilt seems to partially result from a locus on linkage group G causing a proportional decrease in resistance to colonization of the tap root by *Fusarium* as well as linkage groups C2, G and N that reduced foliar symptoms but not root infection [102,103]. SDS QTLs on linkage group G displayed a robust allelic difference in NIL populaces resulting from cultivars Essex and Forrest [100,104]. At least six QTLs with two- and multi-way interactions indicate additive gene action that advocate gene pyramiding for durable resistance. The magnitude of disparity elucidated by recognized QTLs was less than the heritability of the character [105], whereas, out of six QTLs responsible for resistance against the disease, four were located on linkage group (LG) G with 50% of the difference in SDS occurrence; two QTLs on LG C2 [106]. LGs G, C2, I and N comprise QTLs for resistance to SDS [107].

The 7.5 ± 0.5 cM area of chromosome 18 (linkage group G) was revealed to comprehend several resistance loci employing recombination actions from four near-isogenic lines population and nine DNA markers [103]. Three to four genes, viz., *QRfs*-, *QRfs*1-, *QRfs*2- and *QRfs*3-rich islands, were inferred to conveyresistance in linkage group G [108]. QTLs for resistance were recognized on linkage group G by BARC-Satt163, linkage group N by BARC-Satt080 and linkage group C2 by BARC-Satt307 [107]. Recombinant inbred lines containing all the three QTLs for resistance were significantly (*p* ≤ 0.05) more resistant than other recombinant inbred lines with environmentally stable presentation. SSR marker Satt183 has been identified as conferring resistance to SDS on molecular linkage group J with a strong effect on the QTL *Rfs*1 [90]. SSR marker Satt183 has been identified as conferring resistance (56% variation) to SDS on molecular linkage group J with an LOD score of 2.53. Data obtained from regression of the disease score revealed that Satt183 had the most robust increase in the QTL *Rfs*1 [90]. The SSR marker Satt183 (1.2 cM) linked with this QTL was used for clustering of soybean genotypes. The markers amplified four alleles and clustered the genotypes in four groups. The first group comprised 22 genotypes including *Glycine soja*; the second 15; the third 19 and the fourth 10 genotypes [61]. Anderson et al. [108] recognized quantitative trait loci underlying resistance to *Fusarium* wilt in MD96-5722 by the ‘Spencer’ recombinant inbred line population. Fourteen QTLs linked to *Fusarium* wilt resistance through a SNP-based linkage map employing PI 438489 are a valuable foundation for resistance in soybean crop improvement. A SNP-based genetic linkage map of soybean identified as *Fusarium* wilt resistant and could be beneficial in breeding programs developing resistant cultivars [109]. Another linkage map based on SNP markers was built to map QTLs for resistance against SDS [110]. These genetic linkage maps were also constructed employing the MD96-57229 ‘Spencer’ recombinant inbred line population [111]. Some other reports are available on identification of QTLs and construction of linkage maps based on SNP markers for SDS [112,113].

One of the important toolboxes of soybean breeders is called Soybase, which includes information on more than 55 QTLs for SDS. These QTLs have been identified in different populations of soybean. It is important to identify maximum numbers of QTLs and linked DNA markers resistant against SDS. These steps are worthwhile for development of SDS-resistant cultivars through marker-assistedselection. Bao et al. [114] applied an association mapping approach to identify loci responsible for variations in resistance against SDS. A total of 1536 SNP markers were used during this study on 282 soybean lines. Two novel loci on chromosomes 3 and 18 were identified. Results of this study demonstrated the utility of association mapping in the identification of important loci in soybean.

#### 2.1.5. Purple Seed Stain

Purple seed stain triggered by *Cercospora kikuchii* (T. Matsu. & Tomoyasu) Gardner diminishes market rating, processing potentials, germination and vigor of seed [115,116,117,118]. This fungus is responsible for causing purple seed stain (PSS) and *Cercospora* leaf blight (CLB) in soybean. PSS causes symptoms on seed pods and seeds; however, CLB causes symptoms on leaves and petioles. These symptoms include an emblematic darkpurple-colored abrasion. This lesion is a result of the production of cercosporin by the pathogen [119]. Identification of resistance against cercosporin is an important topic of research which can be useful for development of cercosporin-resistant host plants. During the identification of self-resistance to cercosporin in this fungus, Callahan et al. [119] observed that the *CFP* gene is responsible for resistance against cercosporin. Further, one of the sensitive fungi, *Cochliobolus heterostrophus*, was transformed with the *CFP* gene and cercosporin resistance was noticed [120]. In an experiment, Chanda [121] tried to identify the genes responsible for cercosporin biosynthesis. During the experiment, he isolated proteins from *C. kikuchii*. Isolated proteins of the cultures grown under light conditions and dark conditions were compared with the application of a proteomic method. Results of the experiment demonstrated upregulation of six proteins and downregulation of two in the cultures grown under light conditions. Purple seed stain is a major limiting factor for profitable marketability, as it adversely affects the excellence of seed [122,123]. In India, varied percentages of yield loss are reported due to purple seed stain by different research groups, i.e., 15–30% [124] and 36–80% [125]. Economic yield losses make it an objectionable disease in the USA [126]. However, few reports are available on significant yield loss of soybean caused by PSS [117,123].

PI 80,837 and SJ2 are the only reported partially resistant sources for PSS [127,128,129,130] under abstemiously robust (F_2_h^2^ = 0.91 and F_3_h^2^ = 0.51) genetic control [127]. Genetic investigations revealed that resistance to *C. kikuchii* seed infection in cultivar PI80837 is causedby a single dominant gene *Rpss*1 on molecular linkage group G [130]. The candidate resistance gene was mapped between Sat_308 (6.6 cM) and Satt594 (11.6 cM) on molecular linkage group G. These markers may be beneficial in marker-aided selection for employing PSS resistance from PI 80,837 in a breeding scheme [131]. Alloatti et al. [132] detected SSR markers linked to resistance against purple seed stain in soybean. During their study, two populations were developed by crossing PI 80,837 (resistant) with AP 350 and MO/PSD-0259 and further used for the identification of SSR markers linked to purple seed stain resistance genes in soybean. Two SSR molecular markers, Satt115 and Satt340, have been shownto be linked with resistance genes in both populaces.

#### 2.1.6. Cercospora Leaf Blight

*Cercospora* leaf blight (CLB) is one of the foliar fungal diseases of soybean caused by *Cercospora kukuchii*. Initially, *C. kikuchii* was considered as the only causal agent of CLB [133]. However, some recent research findings include some other *Cercospora* spp. (*C.* cf. *sigesbeckiae*, *C.* cf. *flagellaris* and *C.* cf. *nicotianae*) as causal agents of CLB [134,135,136]. Due to the initial identification of *C. kikuchii*, this fungus gained more attention from researchers and at present it is a well-studied pathogen among all four identified pathogens of CLB. This disease is a cause of severe yield losses in soybean throughout the world [136].

The symptoms include appearance of purple and brownish spots on leaves. Further, these colors deepen and lead to premature defoliation in soybean. Generally, this disease is managed by fungicide application. However, continuous applications of fungicides may be a cause of development of fungicide resistance [137]. Therefore, it is important to develop and utilize CBL-resistant soybean varieties to overcome yield losses due to *C. kikuchii*. Purple seed stain (PSS) and *Cercospora* leaf blight are both caused by the same fungus, *C. kikuchii.* According to some previous reports, genetic resistance against purple seed stain has been identified in few soybean cultivars [129,130,131]. Consequently, it is also noticed that there is no correlation between resistance against PSS and CLB [138,139]. Very little is known about resistant against CLB pathogens in soybean. However, a recent study was conducted by Kashiwa et al. [136] based on a high-throughput detached leaf inoculation screening method for selection of CLB-resistant soybean cultivars. One CLB-resistant soybean cultivar (WC54) was identified in Argentina.

#### 2.1.7. Anthracnose

Anthracnose is a major disease of soybean caused by the fungal pathogen *Colletotrichum truncatum* (Schw.) Andrus & W.D. Moore [140]. Estimated yield loss due to this fungal pathogen in soybeans is 25.4 million tons in countries including Argentina, Bolivia, Brazil, Canada, China, India, Paraguay and the USA [141]. The effects of this disease on yield reduction in soybean have been observed in central as well as southern states of India [142]. According to Sharma et al. [143], India faced about 16–25% soybean yield loss due to anthracnose disease. *C. truncatum* has been considered as the main causal agent of anthracnose disease in soybean, however, some other species are also recognized as causal agents of anthracnose. This may be due to the undergoing changes in the genus [144].

Identification of the sources of genetic resistance against this disease may be a possible solution to combat yield loss [145]. Attempts at screening of soybean germplasm for the identification of anthracnose resistance have been carried outby various research groups [146,147]. Recently, Nataraj et al. [142] screened a total of 225 soybean genotypes and five genotypes, viz., EC 538828, EC 34372, EC 457254, AKSS 67 and Karune, and showed their response as highly resistant. Reports are lacking on inheritance of anthracnose resistance which is an important step towards resistance breeding for this disease. According to Nataraj et al. [142], anthracnose resistance in soybean is governed by two major genes interacting in a complementary fashion. Apart from morphological screening, molecular marker-based screening has also been carried out in soybean for the identification of anthracnose-resistant genotypes. In a study, Sajeesh et al. [148] recognized DSb 12 as an anthracnose-resistant genotype on the basis of molecular marker analysis. Furthermore, Kumawat et al. [149] used SSR markers for screening of soybean against anthracnose and, as a result, the EC457,254 genotype was identified as a source of resistance against anthracnose disease.

#### 2.1.8. Brown Stem Rot

Brown stem rot (BSR) is a major disease of soybeans caused by the soilborne fungus *Cadophora gregata* [150], formerly *Phialophora gregataf*. sp. sojae [151]. The fungus inhibits the movement of water and nutrients in the stem of soybean plants which are needed for normal growth and development. Most of the instances of BSR disease can be recognized only after full pod development [152,153,154]. This disease is generally diagnosed as nutrient deficiency. To overcome misidentification of BSR, recently, McCabe and Graham [155] proposed a diagnostic approach based on genes and their network for fast and accurate identification. This approach may be helpful in management of this disease. About a 38% yield loss in soybean crops has been reported due to BSR [154].

Identification of genetic resistance to BSR is a significant and effective approach to decrease yield reduction. Several research groups have used marker-assisted breeding to map BSR resistance genes in soybean. Initially, RFLP markers were used by Lewers et al. [156] to map the *Rbs*3 gene. Furthermore, Klos et al. [157] applied SSR markers to authenticate the genomic region of the *Rbs*3 gene recognized by Lewers et al. [156]. Bachman et al. [158] applied SSR markers for mapping of *Rbs*1 and *Rbs*2 genes in soybean. These three genes were mapped on chromosome 16 of soybean. Moreover, Perez et al. [159] mapped novel sources of BSR resistance on chromosome 16.

When mapping BSR resistance in soybean, Rincker et al. [160] identified different soybean cultivars, i.e., PI 84946-2, PI 437833, PI 437970, L84-5873 and PI 86150, as BSR resistant. Further, the resistance was mapped to intervals ranging from 0.34 to 0.04 Mb, inclusive of BARCSOYSSR_16_1114 and BARCSOYSSR_16_1115 on chromosome 16. On the basis of these findings, it was concluded that only one gene is responsible for BSR resistance in soybean [155].

#### 2.1.9. Powdery Mildew

Powdery mildew, a common leaf disease, is a usual biotic factor responsible for yield reduction in soybean. Powdery mildew is caused by the fungus *Microsphaera diffusa* Cooke & Peck [161]. The main symptoms of this disease are a white, powdery coating on infected leaves of soybean. Due to this coating, the photosynthesis rate is reduced more than 50% [162]. Initially, the powdery patches are seen on the leaves but, after a few days, they cover the entire surface of the leaf. Severity of disease depends upon the genetic character of the cultivars. After severe infestations, infected plants may defoliate hastily [163]. Reduced pod filling is a major result of severe infection in soybean. This indicates that powdery mildew infection is not only responsible for a yield reduction of about 35% [164] but it also affects the quality of seeds in soybean. Both of these reasons are major causes of economic loss for soybean growers. These problems necessitate the development of powdery mildew-resistant soybean cultivars. Recently, Dunn and Gaynor [165] conducted a field experiment in Australia and identified Djakal as a highly resistant soybean cultivar against powdery mildew disease.

Initial studies advocated the Mendelian way of inheritance of powdery mildew resistance in soybean [166,167]. According to earlier reports, the resistance to PMD may be classified as all-stage resistance (ASR) and adultplant resistance (APR). This indicates the expression of resistance according to the growth stage of the plant [168,169] and expression of resistance genes. The studies conducted on identification of inheritance of host plant resistance to powdery mildew (PMD) confirm that the *Rmd* locus has three alleles (*Rmd*, *Rmd*-c and *rmd*) [170]. PMD resistance in adult plants has been reported to be controlled by a single gene, *Rmd* [167,171], however, Lohnes and Bernard [170] reported the role of *Rmd*-c in providing PMD resistance from germination to maturity of soybean crops. According to Kang and Mian [172], a single dominant gene has a role in PMD resistance in the soybean cultivar PI 243540 during all stages of soybean plant growth. During their study, the gene Rmd_PI 243540 from the cultivar PI 243540 was found to be located between SSR marker Sat_224 and SNP marker BARC-021875-04228. Both markers were linked with the PMD resistance gene *Rmd* with a distance of 9.6 and 1.3 cM, respectively.

During a study conducted for genetic mapping of powdery mildew resistance genes and identification of gene linked markers in soybean, a total of 334 F7 RILs were developed from Wyandot (susceptible) and PI 567301B (resistant) cultivars. The developed genetic map showed PMD resistance was flanked by two SSR markers within a 3.3 cM interval on chromosome 16. Furthermore, SNP and CAPS markers with an *Rsa*1 recognition site were also applied in the study. Both of these markers were also found in the same location of the genetic map [173].

Various attempts have been carried out on the use of molecular markers for molecular characterization and diversity analysis among soybean genotypes for powdery mildew resistance. De More et al. [174] used SSR analysis in an F2 population of crosses MGBR95-20937 × IAC-Foscarin 31 and MGBR-46 × EMBRAPA 48 to identify PMD resistance gene-linked markers. In their study, Sat_366 and Sat_393 were found to be located 9.41 and 12.45 cM from PMD resistance genes. Furthermore, Linh et al. [175] conducted an experiment on 36 soybean genotypes with 14 SSR markers and studied genotypes were grouped according to their reactions with powdery mildew disease. PMD-resistant genotypes were grouped distantly from susceptible soybean genotypes in the study. In a recent study, Zhou et al. [169] developed RIL populations based on crossing between Zhonghuang 24 (ZH24) and Huaxia 3 (HX3) and used them to analyze adultplant resistance (APR) to PMD in soybean. The results revealed that PMD resistance was governed by a single dominant locus.

#### 2.1.10. Frogeye Leaf Spot

Frogeye leaf spot (FLS) is a disease of soybean caused by the fungal pathogen *Cercospora sojina* Hara [176]. It is also known as *Cercospora* leaf spot [177]. About 60% yield losses are reported owing to FLS in soybean [178]. This fungal pathogen can survive on crop residue, however, in some cases the infection may be seedborne. FLS can occur at any growth stage of soybean plants [179]. The symptoms of FLS mostly appear on the leaves, however, sometimes these symptoms can be seen on the stems, pods and seeds.

Identification of host resistance in soybean against FLS is the best way to manage this disease, due to development of fungicide-resistant strains of *C. sojina* [180,181]. According to research findings, currently, three genes, viz., *Rcs*1, *Rcs*2, and *Rcs*3, have been found as sources of resistance against FLS in soybean. Occurrence of several races of the pathogen [182] is a big challenge in the development of FLS-resistant varieties of soybean. To develop FLS-resistant soybean varieties, the first step is to identify FLS-resistant soybean cultivars and the locus of the resistance gene in the genome. Recently, Smith [183] worked on two RIL populations for the identification of QTLs responsible for FLS resistance in soybean. In their study, two QTLs responsible for FLS resistance were detected. These QTLs may be helpful in the development of FLS-resistant soybean varieties. Consequently, Mishra et al. [184] identified eleven soybean genotypes, i.e., vs. 2004–9, vs. 2005–40, vs. 2006–17, DSB 11, NRC 84, AMS-MB-5-19, VLS 86, MACS 1407, MACS 1442, NRC 88 and Himso 1685, as FLS resistant. Furthermore, in another study SNP markers were used on an ‘Essex’ × ‘Forrest’-based RIL population for identification of QTLs for FLS resistance. Two QTLs were identified on chromosome 13 and chromosome 19, respectively [185]. The finding of this study may be useful in understanding the mechanism of resistance against *C. sojina*. In a previous study, Sharma and Lightfoot [186] identified two SSR markers, i.e., Satt319 on LG C2 (chromosome 7) and Satt632 on LG A2 (chromosome 8), linked to the QTL responsible for FLS resistance in soybean.

Earlier, some studies based on field as well laboratory research were carried out for the identification of FLS-resistant soybean lines. Mengistu et al. [187] conducted an experiment for this purpose and identified PI 437726, PI 438302B and PI 494851 lines as sources of the *Rcs*3 gene. The study conducted by Mian et al. [188] on FLS race 11 also concluded with the recognition of LN 97-15076 and S99-2281 lines of soybean as sources of the *Rcs*3 gene. According to the authors, these soybean lines may be utilized in developing FLS-resistant cultivars. In another investigation, Pham et al. [189] characterized various soybean genotypes for FLS resistance and identified two genotypes, namely, PI 594891 and PI 594774, as sources of FLS resistance with the presence of the *Rcs*3 gene.

#### 2.1.11. White Mold

White mold in soybean is caused by *Sclerotinia sclerotiorum* (Lib) de Barry, a fungal pathogen. Soybean white mold (SWM) disease is also known as *Sclerotinia* stem rot. The SWM disease is reported as the fourth most important cause of yield reduction in the United States [190]. Host plant resistance is an important method of management of SWM disease. Under field conditions, various physiological factors play major roles in determination of SWM resistance in soybean crops [191]. Applications of molecular markers in association with field experiments have opened new windows due to non-dependency of molecular markers under environmental conditions. For SWM resistance gene as well as QTL identification, Kim and Diers [192] conducted an experiment and identified two QTLs responsible for resistance against SWM. However, both of the identified QTLs were unable to express real resistance and after further evaluation these QTLs were recognized with their role in white mold disease avoidance. Moreover, soybean genotypes also demonstrate their resistance ability according to the effectiveness of races of *S. sclerotiorum* [193].

Examples of various studies on unraveling the mechanism of SWM resistance in soybean are available. Kim and Diers [192] identified three, Aradhana et al. [194] twenty-eight, Guo et al. [195] seven, Vuong et al. [196] four and Huynh et al. [197] three QTLs while studying soybean for detection of sources of SWM resistance. Applications of advanced molecular approaches for SWM resistance have also been carried out. Consequently, association of SNP markers with SWM resistance was observed by various researchers [198,199,200,201,202]. In a study conducted by Moellers et al. [203], 58 SNP-based loci with main effects and some others with epistatic effects were reported to be linked with SWM resistance. Consequently, Boudhrioua et al. [204] carried out genome-wide association mapping-based analysis and reported a new QTL located at chromosome 1 linked with white mold resistance in soybean.

### 2.2. Oomycete Diseases

#### *Phytophthora* Rot

*Phytophthora* root and stem rot diseases are produced by the soilborne pathogen *Phytophthora sojae*. This disease may cause severe damage in flooded areas and damages soybean crops throughout the year [205]. Generally, *Phytophthora* root rot causes yield loss of about 35–40% but in severe conditions it may even causes 100% yield loss. Development of resistant cultivars is the most appropriate tactic for regulating this disease.

There are two mechanisms of resistance to *P. sojae* that have been observed in soybean, i.e., partial and complete resistance [206]. The complete resistance is governed by a single gene (*Rps*) and it is considered as the best mechanism of resistance owing to its race-specific activity. However, partial resistance has limitations under higher infection of *P. sojae* which may be due to the involvements of many genes with their activities against colonization and spread of the pathogen [207].

After development of the linkage map of soybean [46], advancement has been made towards mapping of molecular markers on linkage groups. Mappings of different genes responsible for resistance against *P. sojae* have been reported (Table 1). The *Rps*1, *Rps*2, *Rps*3, *Rps*4, *Rps*5, *Rps*6, *Rps*7 and *Rps*8 loci have been mapped on N, J, F, G, G, G, N and F linkage groups, respectively [46,208,209,210,211,212,213]. Some researchers reported the mapping of *Rps*4 and *Rps*8 loci close to *Rps*6 and *Rps*3, correspondingly [210,212]. One RFLP marker, pT-5, was reported as a linked marker to the *Rps*5 gene [214]. However, Demirbas et al. [210] did not find any SSR marker linked to this gene. According to the researchers, the *Rps*1 locus hasfive functional alleles, which may be a reason for its control efficiency against *P. sojae*. In total, 33 *Rps* genes have been reported to date [204,215]. Amongst all these genes, only six have been observed in commercial cultivars of soybean and provide complete resistance against *P. sojae*. Apart from these genes, soybean contains some of the genes responsible for partial resistance as well [14]. Various soybean lines/genotypes have been identified as well as developed as source of resistance against different biotic stresses (Table 2).

### 2.3. Bacterial Diseases

#### 2.3.1. Bacterial Blight

Bacterial blight is one of the most common diseases of soybeans caused by *Pseudomonas syringae* pathovar glycinea (Psg). It has been established for many years [228]. Soybean cultivars resistant to the causal agent of bacterial blight exhibit a hypersensitive (necrosis) reaction (HR) to infection [229,230,231]. Ashfield et al. [232] carried outan experiment on soybean and identified *Rpg*1 as a source of resistance against bacterial blight disease in soybean. The gene was mapped on linkage group F, linked by the markers K644 and B212. The RFLP markers R45, php2265 and php2385 were also found to be co-segregated with this gene. According to Singh [233], the *Rpg*1b gene (a type of nucleotide-binding site leucine-rich repeat (NBS-LRR) gene) is responsible for resistance against BLB in soybean.

#### 2.3.2. Bacterial Pustule

Bacterial leaf pustule (BLP), caused by *Xanthornonas axonopodis* pv. glycines (Xag), is a worldwide disease of soybean, particularly in warm and humid regions. A novel *Xanthomonas vasicola* strain was isolated from the leaves of soybean plants infected with bacterial blight under field conditions [234]. To date, little is known about molecular mechanisms of BLP resistance. To achieve yield stability, disease resistance breeding is the most important and accurate approach [235,236]. Initially, a single gene, *rxp*, was identified as a source of BLP resistance [237,238] which was detected in the soybean cultivar PI 71569. It was also observed that *rxp* provides resistance against wildfire disease in soybean [239]. The dominant *Rxp* allele, which confers susceptibility to the BLP disease, was mapped 3.9 cM from Satt372 and 12.4 cM from Satt014 on Chr 17 [240]. The recessive *rxp* was also located between SNUSSR17_9 and SNUSNP17_12 on Chr 17 using a QTL fine mapping approach. A genome-wide association mapping approach was applied to identify one significant SNP associated with BLP resistance [241]. An RNA-Seq analysis was carried out to identify differentially expressed genes (DEGs) between resistant and susceptible soybean varieties under the infection of the Xag isolate [242]. Several minor QTLs with resistance against various Xag isolates have been mapped after use of recombinant inbred line (RIL) populations developed using ‘Suwonl57′ and ‘Danbaekkong’ soybean cultivars as resistant and susceptible parents, respectively [243]. In another experiment, a BLP resistance gene/QTL was reported to be located near (21.5 cM) to the SSR marker Sat_108 on Chr 10 [241,242]. In a recent study, Wang et al. [244] identified the role of GmHPL in Xag resistance in soybean.

Using a linkage mapping in a recombinant inbred line (RIL) population, Zhao et al. [245] identified that QTL qrxp–17–2 accounted for 74.33% of the total phenotypic variations. Real-time RT-PCR analysis of the relative expression levels of five potential candidate genes in the resistant soybean cultivar W82 following Xag treatment showed that of Glyma.17G086300, which is located in qrxp–17–2. *Glyma.17G086300* was reported as a potential candidate gene for *rxp*. The QTLs identified in their study are useful for marker development for the breeding of Xag-resistant soybean cultivars.

### 2.4. Nematode Diseases

#### Soybean Cyst Nematode

In the last few years, nematode disease incidents have been reported from major soybean-producing countries, including India. Among different nematode diseases, the soybean cyst nematode (SCN) is *Heterodera glycines* Ichinohe. This nematode is the most devastating obligate parasite of the important leguminous crop soybean. On average, about 30% yield loss has been reported due to this nematode disease in soybean [246]. One major drawback with this nematode disease is that the infestation can only be identified at an advanced stage and before that it causes major damage to the crop [247].

Efforts have been madefor the identification of the source of resistance as well as mapping of SCN resistance in soybean. Initially, Ross and Brim [230] reported the Rhg locus to be responsible for resistance against SCN in soybean. SCN resistance in soybean has been reported to be multi-genic by various researchers [230,231]. Various QTLs have also been mapped for SCN resistance in soybean. Kim et al. [247] mapped the *rhg*1 locus at chromosome 18 and predicted that Satt309 is linked closely with the *rhg*1 locus and it may be applied for marker-assisted selection for SCN resistance in soybean. In a recent study, Vuong et al. [248] applied Infinium SoySNP6K BeadChips and the genotype-by-sequencing (GBS) method for identification of QTL(s) linked to SCN resistance. In their study, they identified QTLs on chromosomes 10 and 18. Zhang et al. [221] reported three SCN resistance loci after GWA analysis of 282 soybean accessions. Recently, Kofsky et al. [222] reported wild soybean (*Glycine soja*) as a source of SCN resistance and also advocated for further research on biochemical mechanism-based analysis of resistance for clarification.

Applications of multi-omics techniques in fast and accurate identification of resistant loci and linkage of molecular markers, metabolites as well as proteins are also important for molecular breeders. Various examples are available on the applications these advanced approaches in the improvement of soybean crop (Table 3).

These techniques played a major role in soybean breeding [260] and made it possible to understand the current drawbacks and possibilities for future research. Transcriptomic-, proteomic- and metabolomic-based studies have their significance in exploration of molecular mechanisms of resistance against different biotic stresses in soybean (Table 3). These techniques are also helpful in the discovery of resistance genes to hasten the breeding process in soybean for the development of new cultivars with desirable traits [261,262].

Advances in plant breeding techniques opened multiple windows for the modification of the targeted genome site of desired genotypes. In this way, zinc-finger nucleases, transcription activator-like effector nucleases and clustered regularly interspaced short palindromic repeats (CRISPR/*Cas*9) have emerged with their strong efficiency in genome modification of soybean crops. However, authentic information on development of biotic stress-resistant soybean varieties is not available. Apart from the currently available molecular breeding technique, the advanced genome editing platforms may be utilized to combat biotic stresses in soybean crops asthese new techniques are efficient in the improvement of soybean through functional characterization of targeted genes and accurate genome editing for desired trait improvement.

## Figures and Tables

**Table 1 plants-11-01967-t001:** Genes/markers reported to be linked with soybean resistance against different biotic stresses.

Locus	Cultivar	Interval	Linkage Group/Chromosome	Franking Markers	Reference
**Soybean rust**					
*Rpp*1	PI 200492	~1 cM	G/18	Sct_187 Sat_064 Sat_117 Sat_372 Satt191	[25,26,27,28,40,42,145]
*Rpp*2	PI 230970		C2/16	Sat_255 Satt620 Satt529	[27,29,46,51]
*Rpp*3	PI 462312		J/16	Satt134 Satt460	[29,30,31,42]
*Rpp*4	PI 459025B	1.9 cM 2.8 cM	C4/18 G/18	Satt288 Satt503, Satt612, Satt288, Satt517, Sat_143, Sct_199, Satt472, Satt191	[33,146,147,148,149,150]
*Rpp*5	PI 200456		N/3	Sat_275 Sat_280 Satt080, Satt125, Satt485, Satt393	[48,127,128]
*Rpp*6	PI 567102B		G/18	Satt324, Satt394	[35,129,130,131]
*Rpp*7	PI 605823		L/19	GSM0547, GSM0548	[132]
*Rpp*1b	PI 594538A	1 cM of *Rpp*1 and ZM01-1	G/18		[36]
***Rhizoctonia* root rot**					
	PI 442031		A2 C2 M	Satt177 Satt281 Satt245	[56]
***Fusarium* wilt**					
		7.5 ± 0.5 cM	G		[103]
	QTL *Rfs*1		J	Satt183	[108]
**Purple seed stain**					
*Rpss*1	PI 80837	6.6 cM 11.6 cM	G/18	Sat_308 Satt594	[132]
*Rpss*1	PI 80837	44.1 cM 46.8 cM	18	Satt115 Satt340	[133]
***Phytopthora* rot**					
*Rps*1a	L88-8470	1.2 cM	N/3	Satt159	[45,151,214]
*Rps*1b	L77-1863		N/3		[152,210]
*Rps*1c	L75-3735		N/3		[152,210]
*Rps*1d	PI 103091	5.7 cM 11.5 cM	N/3	Sat_186 Satt152	[153,154]
*Rps*1k	L77-1794		N/3		[155]
*Rps*2	L76-1988	0.5 cM	J/16	Sat_393	[156]
*Rps*3	L83-570	0.1 cM	F/13	Sat_317	[152,205]
*Rps*4	L85-2352	4.3 cM	G/16	Sat_004	[157,205]
*Rps*5	L85-3059		G/16		[158]
*Rps*6	L89-1581	0.4 cM 0.3 cM	G/16	Sct-187 Sat_372	[159,206]
*Rps*7	L93-3258	6.7cM	N/3	Satt152	[160,206]
*Rps*8	PI 399073	4.0 cM	F/13	Satt154	[161,206]
*RpsZS*18	Zaoshu18	0.9 cM 0.5 cM	2	ZCSSR33 ZCSSR46	[205]
**Bacterial blight**					
*Rpg*1	BSR101		F/13	K644, B212	[175]
**Powdery mildew**					
*PMD*	BRS135	7.1 cM 4.6 cM	16	GMES6959 Satt_393	[216]
*PMD*	PI 567301B	3.3 cM	16	BARCSOYSSR_16_1291	[173]
*Rmd*	PI 243540	9.6 cM 1.3 cM	16	Sat_224 BARC-021875-04228	[172]
*ASR*	ZH24		16	Gm16_428	[169]
**Soybean cyst nematode**					
*Rhg*4	PI 438489B		8	Satt632	[217]
*qSCNPL*10	Pingliang ZDD 11047		10		[218]
*rhg*1	Toyomusume		18	Satt038, Satt309	[219]
*cqSCN*10	PI 567305		10	Satt592, Satt331, and Sat_274	[220]
*rhg*1	PI 438489B		18	Satt309, Sat_168	[221]

**Table 2 plants-11-01967-t002:** List of identified/developed soybean genotypes/lines with resistance against various diseases.

Genotype/Line	Disease	Reference
SRE-Z-11A, SRE-Z-11B, SRE-Z-15A	Soybean rust	[23]
PI 441001	Soybean rust	[24]
USP 97-08135	Soybean rust	[25]
PI 416764, PI 462312, KS 1034	Soybean rust	[26]
TGx 1993 4FN, TGx 1995 5FN, PI 594538A	Soybean rust	[27]
PI 594723, PI 594538A, PI 587880A, PI 230970, PI 459025A	Soybean rust	[28]
PI 200492	Soybean rust	[26]
PI 230970	Soybean rust	[27]
PI 462312	Soybean rust	[30,31]
PI 459025B	Soybean rust	[33]
PI 200456	Soybean rust	[127]
PI 567102B	Soybean rust	[35]
PI 605823	Soybean rust	[132]
PI 594538A	Soybean rust	[36]
AGS-129, G00056	Rhizoctonia root rot	[222]
PI 442031	Rhizoctonia root rot	[56]
PI 417361, PI 504488, PI 88490, PI 346308, PI 416779, PI 417567, PI 381659, PI 417567, PI 407749	Purple seed stain	[223]
PI 80837	Purple seed stain	[132]
L88-8470	*Phytophthora* rot	[151]
L77-1863, L75-3735, L83-570	*Phytophthora* rot	[151]
PI 103091	*Phytophthora* rot	[152,153]
L77-1794	*Phytophthora* rot	[154]
L76-1988	*Phytophthora* rot	[155]
L85-2352	*Phytophthora* rot	[156]
L85-3059	*Phytophthora* rot	[157]
L89-1581	*Phytophthora* rot	[158]
L93-3258	*Phytophthora* rot	[159]
PI 399073	*Phytophthora* rot	[160]
Zaoshu18	*Phytophthora* rot	[203]
E00003	*Phytophthora* rot	[224]
JS 20-98, JS 20-34, MAUS 162	Charcoal rot	[225]
BSR101	Bacterial blight	[226]
TGX1987-62F, TGX1990-15F	Bacterial pustule	[227]
Djakl	Powdery mildew	[165]
PI 567301B	Powdery mildew	[173]
PI 243540	Powdery mildew	[169]
ZH24	Powdery mildew	[172]
PI 438489B	Soybean cyst nematode	[221]
Pingliang ZDD 11,047	Soybean cyst nematode	[218]
Toyomusume	Soybean cyst nematode	[219]
PI 567305	Soybean cyst nematode	[220]

**Table 3 plants-11-01967-t003:** Advanced approaches and their application in biotic stress resistance in soybean.

Disease	Approach	Results	Reference
Soybean cyst nematode	Transcriptomics	6792 gene transcripts	[249]
Soybean cyst nematode	Transcriptomics	6000 variable genes	[250]
Soybean cyst nematode	Transcriptomics	3746 DEGs	[251]
Soybean cyst nematode	Proteomics	456 differentially expressed proteins	[252]
Soybean cyst nematode	Metabolomics	14 specific differential metabolites	[253]
Asian soybean rust	Proteomics	Distinct protein	[254]
Asian soybean rust	Proteomics	70 differentially expressed proteins	[255]
Asian soybean rust	Metabolomics	Defense secondary metabolite production	[256]
*Phytophthora* root rot	Proteomics	46 differentially expressed proteins	[257]
*Phytophthora* root rot	Metabolomics	90 differentially accumulated metabolites	[216]
Downy mildew	Transcriptomics	52 differentially expressed genes	[258]
Charcoal rot	Transcriptomics	1219 DEGs	[259]

## Data Availability

Not applicable.

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
