# Peer review of "Molecular Breeding to Overcome Biotic Stresses in Soybean: Update"

_plants, 2022, doi:10.3390/plants11151967_

Round 1

Reviewer 1 Report

This is a difficult review to read. English language and style require extensive editing throughout the whole manuscript. In many sentences, English language and syntax is not used in the correct way, while there are many places where commas are used incorrectly.

The review is not well organized. The authors have done a good job of researching the literature but the problem is that they report data without focus and organization. 

In addition, please add a Table with the resistant soybean cultivars for each pathogen and virus developed through conventional methods, molecular methods as well as the modern molecular techniques.

I also suggest that you add some figures in your review.

Are there any resistant soybean cultivars developed via the use of zinc-finger nucleases and CRISPR/Cas9 that are used in the market (not just for research  purposes)? If yes, how much better are these cultivars compared to the older cultivars? If no, then please state the fact.

Please add more data on the tolerant cultivars developed through these modern techniques, otherwise the review will not include any new piece of information.

Author Response

Response sheet attached

Reviewer 2 Report

Accept in present form

Author Response

Response sheet attached

Reviewer 3 Report

I've reviewed the paper "Molecular breeding to overcome biotic stresses in soybean". The authors provide a comprehensive review of disease resistance in soybean, and this paper will be of interest to soybean researchers. However, several similar reviews are already available. To make this paper even more informative, the following revisions need to be made.

P1 affiliation

The country name is not included in the description of the author's institutional affiliation. Since the text mentions the damage in India, please include the name of the country in the affiliation.

P2 L73

Why isn't Rpp6 included here? Please include Rpp6.

P3 L100

what is variation guesstimates? Please explain in more commonly used terms.

P3 L109

Rpp4C is just a candidate gene name copied directly from the paper, so please check and describe which one is applicable to the annotated genes in the current soybean genome.

P3 L110 

what is intrant genes ?

P3 L112

Rpp4-C4 is Rpp4C4. No spaces or hyphens are required.

P3 L117

What do you mean by stamping markers?

Table1

Only some chromosome numbers are written. If possible, please look up and describe the physical location of the SSR markers as well.

P4 L124.

It is incremental since it is mentioned at the beginning. I think Rpp1-6 are fine, Rpp7 was not sure of its existence, shouldn't Rpp7 be mentioned? 

P4 123-130

Has the Rpp4 disease resistance gene been isolated by applying the current gene annotation? Specifically, it would be difficult to say that the Rpp4 gene has been identified. Please research and describe this area in detail.

P4 L125-127

Please specify what Pedley et al. et al. revealed by gene silencing.

P7 L265

What is Contender genes ?

Please correct several places where gene names are not italicized.

Author Response

Response sheet attached

Round 2

Reviewer 1 Report

Authors have made improvements to their manuscript. 

It is a much better review than the first draft they submitted.